# Sparse Query Dense: Enhancing 3D Object Detection with Pseudo points

## ABSTRACT

Current LiDAR-only 3D detection methods are limited by the sparsity of point clouds. The previous method used pseudo points generated by depth completion to supplement the LiDAR point cloud, but the pseudo points sample process was complex, and the distribution of pseudo points was uneven. Meanwhile, due to the imprecision of depth completion, the pseudo points suffer from noise and local structural ambiguity, which limit the further improvement of detection accuracy. This paper presents SQDNet, a novel framework designed to address these challenges. SQDNet incorporates two key components: the SQD, which achieves sparse-to-dense matching via grid position indices, allowing for rapid sampling of large-scale pseudo points on the dense depth map directly, thus streamlining the data preprocessing pipeline. And use the density of LiDAR points within these grids to alleviate the uneven distribution and noise problems of pseudo points. Meanwhile, the sparse 3D Backbone is designed to capture long-distance dependencies, thereby improving voxel feature extraction and mitigating local structural blur in pseudo points. The experimental results validate the effectiveness of SQD and achieve considerable detection performance for difficult-to-detect instances on the KITTI test.

## CCS CONCEPTS

• **Computing methodologies → Object detection**.

## KEYWORDS

sparse-to-dense matching, sparse query dense, sparse 3D Backbone, local structural blur, point cloud sparsity

### ACM Reference Format:

Anonymous Authors. 2024. Sparse Query Dense: Enhancing 3D Object Detection with Pseudo points. In *Proceedings of the 32nd ACM International Conference on Multimedia (MM'24), October 28-November 1, 2024, Melbourne, Australia.* ACM, New York, NY, USA, 10 pages. https://doi.org/10.1145/nnnnnnn.nnnnnnn

## 1 INTRODUCTION

3D object detection aims to locate and classify objects in 3D space and is a vital perception task that plays a crucial role in autonomous driving [1, 8, 20, 21, 23–25, 27, 32, 46, 51]. The reliance on LiDAR data comes at the cost of point density variations across distances. Other factors such as occlusion play a role, but the primary reason

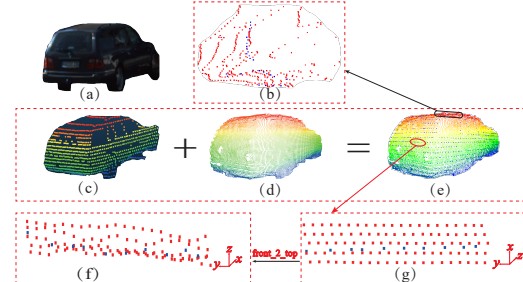

**Figure 1: The local structure distortion and edge noise problems of the pseudo point cloud. (a) is an image of a car, while (c), (d), and (e) show the LiDAR point cloud, pseudo point cloud, and mixed point cloud of (a), respectively. (b) illustrates the edge noise at the top of the object, with irregularly distributed pseudo dots (red) contrasting with LiDAR dots (blue). Fig. 1(g) showcases the point cloud extracted from the white area in Fig. 1(c), indicating that pseudo points roughly parallel the LiDAR lasers from the front view. From Fig. 1(g)'s front to Fig. 1(f)'s top view. The top view in Fig. 1(f) reveals local structural blurring due to positional differences between pseudo and LiDAR points, despite frontal alignment.**

is the natural divergence of points from the LiDAR with increasing distance due to the angular offsets between the LiDAR lasers. Thus, the LiDAR sensor receives fewer measurements of remote objects.

In contrast, an RGB sensor can see hundreds of pixels, which leads to a simple approach: converting image pixels into pseudo points to enrich the LiDAR point cloud [37, 46]. The process begins with depth completion, projecting each image pixel into 3D space using estimated depth to create a pseudo point cloud for the entire scene [33]. Subsequently, sampling specific pseudo points from the pseudo point cloud and fusing them with the LiDAR points increases the point density for objects.

Many studies have been proposed to augment the LiDAR point cloud by integrating pseudo points [37, 38, 46]. VirConv [37] segments preprocessed pseudo points by distance, maintaining all distant pseudo points while randomly selecting a specific number of closer pseudo points to create a new pseudo point cloud for fusion with the LiDAR point cloud. Virconv achieved the SOTA (state-of-the-art) performance on the KITTI Car 3D object detection leaderboard. However, the preprocessing of the pseudo points is time-consuming. The complete process of using pseudo points obtained via depth completion [17, 18, 34] to supplement the LiDAR point cloud is first to project LiDAR points onto the sparse depth map, generating a dense depth map through depth completion, and converting the dense depth map into 3D pseudo points for subsequent sampling. Our method aims to simplify the preprocessing pipeline of pseudo points by directly sampling the needed data

from dense depth maps, reducing reliance on the preprocessing of pseudo points, which helps with the rapid sampling of large-scale pseudo points.

Inaccurate pseudo points frequently fail to depict the delicate local structure of object surfaces accurately and introduce significant noise, particularly around object edges. This issue is illustrated by comparing an image of a car with its corresponding LiDAR (blue dots) and pseudo (red or colored dots) point clouds in Fig. 1(a) and Fig. 1(d), with the mixed point cloud shown in Fig. 1(c). A detailed examination of the car's top area in Fig. 1(b) reveals that the pseudo points at the edges of the object contain a large amount of noise. Fig. 1(g) showcases the mixed point cloud extracted from the white area in Fig. 1(c), indicating that pseudo points roughly parallel the LiDAR lasers from the front view. To further highlight local structural ambiguities, transitioning Fig. 1(g) to the top view in Fig. 1(f), where it is observable that pseudo points, despite being adjacent and aligned with the LiDAR lasers in the front view, display positional discrepancies compared to the LiDAR points when viewed from the top. Such inaccuracies ultimately limit the further improvement of detection performance, especially in voxel-based detection, where averaging pseudo and LiDAR point coordinates can blur an object's local structure.

Two noteworthy phenomena can also be observed from Fig. 1: (1) LiDAR points are sparse or entirely absent near the noise pseudo points along the object's edges; (2) despite the local structural ambiguity of pseudo points on an object's surface, they can still globally capture the object's structure, as depicted in Fig. 1(d).

For ease of expression, the LiDAR and pseudo points mentioned in this paper are non-empty pixels on sparse and dense depth maps, and they correspond one-to-one with the LiDAR and pseudo points in the LiDAR coordinate system. In the final stage of SQD, we convert the coordinates of the pseudo points queried in the 2D dense depth map into 3D coordinates in the LiDAR coordinate system.

To address these challenges, we present the SQDNet. SQDNet incorporates two key components: the SQD (Sparse Query Dense) and a novel sparse 3D Backbone. SQD leverages sparse LiDAR points to query dense pseudo points, filling in structural information on object surfaces autonomously. By mapping both sparse LiDAR and dense pseudo points onto 2D-occupied grid maps, SQD achieves efficient sparse-to-dense matching via grid position indices, reducing the computational cost of using LiDAR points to query pseudo points and achieving fast sampling of large-scale pseudo points. Only pseudo points in "occupied" grids of LiDAR are considered, alleviating noise around object edges by ignoring pseudo points in "free" grids of LiDAR. Furthermore, the SQD further queries pseudo points based on the density of LiDAR points within these grids to correct the uneven distribution of pseudo points. The combined point cloud, a mixture of LiDAR and selected pseudo points, undergoes voxelization, creating numerous pseudo voxels around LiDAR voxels due to the pseudo points' relative abundance. Therefore, we design a sparse 3D Backbone that expands the receptive field of features, allowing pseudo voxels to integrate accurate LiDAR voxel features during the convolution process, thereby reducing the impact of inaccuracies from local pseudo points.

Our key contributions are:

- By mapping sparse and dense points to 2D-occupied grid maps, we facilitate rapid sparse-to-dense matching leveraging grid position indices and reduce the computational cost of using LiDAR points to query pseudo points;
- SQD samples pseudo points based on the occupancy status of grids and density of LiDAR points within these grids to alleviate edge noise and the problem of uneven distribution;
- A novel sparse 3D Backbone is designed to model long-range dependencies, alleviating the problem of local structural ambiguity;
- Experimental results demonstrate that using pseudo points to supplement surface structure details of difficult-to-detect objects significantly enhances the model's ability to detect these objects.

## 2 RELATED WORK

### 2.1 Point cloud sample

PointNet [6] uniformly samples 1024 points on mesh faces according to the face area and normalizes them into a unit sphere for 3D object detection. FPS (farthest point sampling) randomly selects a point as a seed point from the original point cloud, and other points are chosen according to Euclidean distance, aiming to maintain structural information [26, 28, 29]. However, FPS tends to choose remote points to cover the entire scene better, which could make downsample points involve excessive irrelevant background points like points on the ground [10]. Yang et al. [45] propose the F-FPS, which effectively preserves the inliers of various instances. Chen et al. [7] select foreground points by prioritizing those with higher semantic scores from the segmentation module and their coordinates. Zhang et al. [49] propose two learnable, task-oriented, instance-aware downsampling strategies for the hierarchical selection of foreground attractions belonging to objects of interest.

In other tasks of point cloud processing, e.g., semantic segmentation, sampling of point clouds is usually also required. IDIS (Inverse Density Importance Sampling) reorders all N points according to the density of each point and selects the top $K$ points [13]. CRS (Continuous Relaxation based Sampling) [2, 44] uses the reparameterization trick to relax the sampling operation to a continuous domain for end-to-end training, and each sampled point is learned based on a weighted sum over the entire point clouds. PGS (Policy Gradient-based Sampling) formulates the sampling as a Markov decision process [41] and learns a probability distribution to sample the points sequentially.

### 2.2 Sparse 3D Backbone

Voxel-based 3D object detection methods convert point clouds into voxels and apply sparse 3D Backbones for feature extraction [12, 19, 28]. Sparse 3D Backbones resemble 2D CNNs in structures, including several feature extract stages and down-sampling operations, typically consisting of regular and submanifold sparse convolutions [14].

Liu et al. [23] introduce amplitude-based sampling modules—SPS Conv (spatial pruned sparse convolution), SPSS Conv (spatial pruned submanifold sparse convolution), and SPRS Conv (spatial pruned regular sparse convolution) to dynamically prune spatial redundancy in data and models effectively without affecting performance,

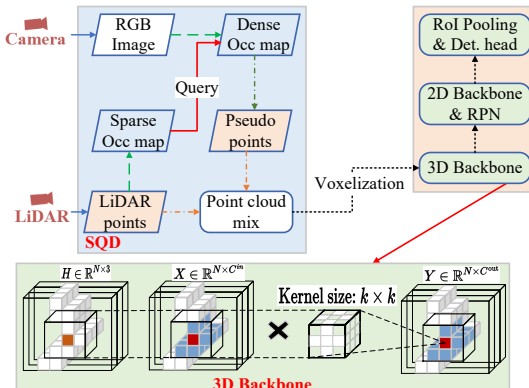

Figure 2: The pipeline of SQDNet. Firstly, the LiDAR point cloud is converted into a sparse depth map via an extrinsic matrix. Subsequently, a depth completion network is used to complete the sparse depth map and generate a dense depth map. Both sparse and dense depth maps are then transformed into 2D-occupied grid maps, allowing for the efficient querying of pseudo points based on grid occupancy and LiDAR point density within these grids. These pseudo points are mixed with LiDAR points to create an enriched point cloud, which is processed by the proposed sparse 3D Backbone for voxel feature extraction and long-range dependency modeling. The pipeline also includes a 2D Backbone, a Region Proposal Network (RPN), and a detection head.

leveraging feature amplitude information. Chen et al. [8] present the Focals-Conv, a dynamic approach for learning feature spatial sparsity by assessing voxel importance through an extra convolutional layer, thereby focusing the learning on more valuable foreground data.

## 3 SQDNET FOR 3D OBJECT DETECION

Fig. 2 illustrates the pipeline of our model, which includes: (1) an SQD block that leverages sparse LiDAR points to query dense pseudo points, autonomously enriching structural details on object surfaces and facilitating efficient sampling of large-scale pseudo points directly from the dense depth map; (2) a sparse 3D Backbone designed for voxel feature extraction and modeling of long-range dependencies, which helps to mitigate the local structural ambiguities inherent in pseudo points. We detail our method in the following sections.

### 3.1 Sparse Query Dense

In SLAM (Simultaneous Localization and Mapping) [4], 2D grid occupancy models the environment by dividing it into 2D-occupied grid maps, with grids marked "occupied" or "free" to denote obstacles. This paper extends the grid occupancy concept for 3D detection, designating grids containing points as "occupied" while those without as "free". We project both sparse LiDAR points and dense pseudo points onto 2D-occupied grid maps and leverage the position indices of grids to facilitate efficient and indirect sparse-to-dense matching.

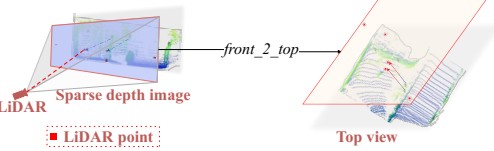

Figure 3: The LiDAR point cloud is projected onto a 2D sparse depth image using an extrinsic matrix, converting it from 3D space to a front view and then further transformed to a top view through coordinate conversion.

To construct 2D-occupied grid maps, LiDAR points are initially converted into a sparse depth map according to the current sensor attitude, as depicted in Fig. 3. In the absence of calibration noise, the projection from the 3D LiDAR coordinate into a 2D image coordinate involves a transformation from the LiDAR measurement to the camera frame and a perspective projection from the camera frame into image coordinates:

$$\mathbf{I}_{\text{sparse}} = \mathbf{P}_{\text{rect}}^{(i)} \mathbf{R}_{\text{rect}}^{(i)} \mathbf{T}_{\text{cam}}^{\text{velo}} \mathbf{X} \tag{1}$$

Where $\mathbf{X} = \{(x, y, z, r)_i\}$ represents a LiDAR point cloud, each point is defined by its 3D location $(x, y, z)$ and reflectance $r$. The transformation from the LiDAR coordinate system to the camera coordinate system is facilitated by the extrinsic matrix $\mathbf{T}_{\text{cam}}^{\text{velo}}$, with the camera's corrected rotation matrix denoted as $\mathbf{R}_{\text{rect}}^{(i)}$ and its projection matrix as $\mathbf{P}_{\text{rect}}^{(i)}$. This transformation yields a sparse depth map $\mathbf{I}_{\text{sparse}} = (u, v, d)$, where $(u, v)$ is the 2D coordinates of the LiDAR point on the sparse depth map, and $d$ represents the depth value.

After converting the sparse depth map from the front view to the top view through a coordinate transformation (changing 2d coordinates from $(u, v)$ to $(d, v)$) [5]. Then, the top view is divided into $m \times n$ grids to establish a sparse occupancy map (Sparse Occ map), as shown in Fig. 4. Similarly, a dense occupancy map (Dense Occ map) is created from the dense depth map produced through depth completion. The occupancy status of these 2D grids is determined by marking grids containing LiDAR or pseudo points as "occupied" and all others as "free". We use the position indices of grids to achieve efficient sparse-to-dense matching, converting LiDAR points query pseudo points into grids query grids, reducing the computational complexity of queries, and thus achieving fast sampling on large-scale pseudo points.

We construct the index arrays $\mathbf{S}_{\text{point2Occ}}$ and $\mathbf{D}_{\text{point2Occ}}$, where the position indices of "occupied" grids in the 2D-occupied grid maps serve as keys, and the coordinates of LiDAR and pseudo points mapped to those grids are stored as values, as shown in Fig. 4. It should be emphasized that the number of points mapped into each grid is not constant.

Fig. 1(b) reveals that LiDAR points tend to be sparse or completely absent near the noise pseudo points along the object's edges. Consequently, pseudo points within a grid marked as "occupied" in the Dense Occ Map but "free" in the Sparse Occ Map at the same location are deemed noise and consequently discarded. To achieve this, We utilize the position indices of "occupied" in the Sparse Occ Map to query grids at the same position in the Dense Occ Map

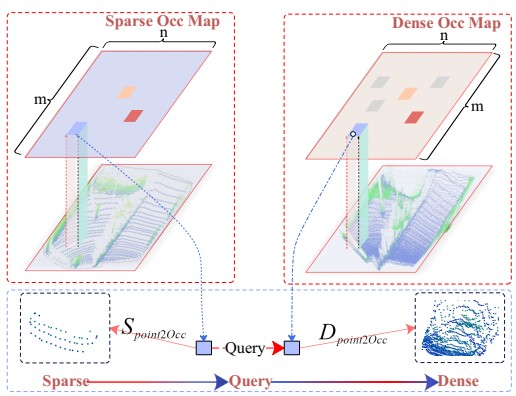

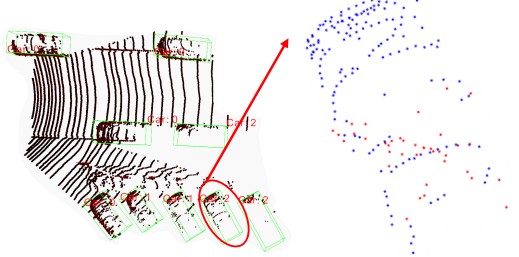

**Figure 4: The pipeline of SQD. Divide the top view of sparse and dense depth maps into $m \times n$ grids to create the Sparse Occ Map and Dense Occ Map, and construct index arrays $S_{point2Occ}$ and $D_{point2Occ}$ to store the correspondence between points and grids. The occupancy status of 2D grids is determined by marking the grid containing points as "occupied". These colored-filled grids indicate the "occupied" grids. SQD leverages grid position indices where the density of LiDAR points falls between $\tau$ and $\mu$ in the Sparse Occ Map to query grids at the same position in the Dense Occ Map. It then uses $D_{point2Occ}$ to query pseudo points mapped to these grids.**

and then preliminarily determine the pseudo points that need to be retained based on $\mathbf{D}_{point2Occ}$.

Due to the uneven distribution of pseudo points generated by depth completion, more pseudo points are mapped into the closer grid, while fewer pseudo points are in the distant grid. To address this, we further query the pseudo points based on the density of LiDAR points within the Sparse Occ Map's grids. The process begins by assigning initial weights to each pseudo point using random numbers uniformly distributed between 0 and 1. If the density of LiDAR points within a grid in the Sparse Occ Map is below the threshold $\tau$, it signifies that the LiDAR points in that area are relatively sparse and require supplementation with pseudo points. Conversely, if the density is below another threshold $\mu$, it indicates that the LiDAR points in that grid are either noise or irrelevant to detection and consequently discarded. The SQD employs grid position indices with densities of LiDAR points between $\tau$ and $\mu$ in the Sparse Occ Map to query grids at the same position in the Dense Occ Map. It then uses $\mathbf{D}_{point2Occ}$ to query pseudo points mapped to these grids. To ensure these critical pseudo points are retained through later stages, SQD assigns them exceptionally high weight values.

Finally, SQD selects pseudo points with weights exceeding the threshold $\sigma$ and converts their coordinates from the 2D dense depth map to 3D coordinates in the LiDAR coordinate system. These pseudo points are then fused with LiDAR points and denote the fused points as $\mathbf{P} = \{p_1, p_2 \cdots p_n\}$, with each point $p_i$ is characterized by coordinates $(x, y, z)$, intensity $\alpha$, and an indicator $\beta$ denoting the point's origin. The intensity of pseudo points is padded by 0.5.

**Figure 5: The figure demonstrates the results of querying pseudo points (red) to supplement LiDAR clouds (blue), with objects marked in green ground truth (GT) boxes. Each box has a red label indicating the object's category and visibility. A point cloud of a hard-category car is specifically chosen and enlarged for closer analysis.**

Finally, we use the proposed sparse 3D Backbone to extract voxel features of fused points.

By mapping LiDAR and pseudo points to 2D-occupied grid maps, we achieve the fast querying of pseudo points, significantly reducing the preprocessing time for pseudo points. Subsequently, based on the density of LiDAR points in the occupied grid, we perform the secondary sampling on the initial queried pseudo points to alleviate the problem of uneven distribution of pseudo points.

## 3.2 Sparse 3D Backbone

Sparse 3D backbone, including submanifold and regular sparse convolutions, are widely used for extracting features from precise LiDAR point clouds [48]. However, directly applying these convolutions to extract features from a mixture of pseudo and LiDAR points poses challenges in further improving 3D object detection performance, primarily due to the problem of local structural blurring of pseudo points.

Fig. 5 demonstrates how the SQD queries pseudo points for enriching the LiDAR cloud, with objects labeled within green ground truth (GT) boxes. Each box is marked with a red label indicating the object's category and visibility. A point cloud of a hard-category car is specifically chosen and enlarged for closer analysis. This figure reveals that the pseudo points queried through SQD can supplement the structural information of the object surface.

The blurring of local structure in pseudo points caused by depth completion is mainly due to inaccurate depth value estimation, resulting in a significant positional deviation between LiDAR and pseudo points in the top view, as shown in Fig. 1(f). VirConv [37] extends the receptive field to 2D image space and uses sparse 2D Backbone to extract voxel features in 2D space, implicitly distinguishing noise patterns. However, VirConv [37] requires frequent transform voxels from 3D to 2D in sparse 3D Backbone. Upon rethinking VirConv, we believe that it is equivalent to ignoring inaccurate depth information, allowing voxels that are not adjacent in 3D space but adjacent in 2D image space to achieve feature interaction, thereby expanding the receptive field of features and modeling long-distance dependencies.

Pseudo points, queried by SQD, are merged with LiDAR points to form an enriched point cloud. After voxelization, the enriched point

**Figure 6: The structure of the proposed sparse 3D Backbone.**

cloud generates pseudo voxels around the LiDAR voxels. Consequently, we consider expanding the **kernel size** of the convolutional kernel to increase the receptive field of features. This expansion allows pseudo voxels to interact with accurate features from LiDAR voxels during the convolution process, thereby alleviating the impact of local structural ambiguity of pseudo points.

The structure of the sparse 3D Backbone is depicted in Fig. 6. Similar to Voxel R-CNN [12], the sparse 3D Backbone in SQDNet comprises *Conv input*, *Conv out*, and *layer*[1 − 4]. Distinguishing from Voxel R-CNN, SQDNet stacks 2× large-kernel convolution [9] in *layer* 1, and adopts **kernel sizes** of [7, 5, 5, 3] across *layer*[1 − 4], respectively.

## 4   EXPERIMENTS

### 4.1   Datasets and Evaluation Metrics

KITTI 3D object detection benchmark [11] contains 7,481 LiDAR and 7,518 frames for training and testing, respectively. The training data is split into 3,712 and 3,769 frames for training and validation (val). The detection results on the val and test set are evaluated with the average precision calculated by 40 recall positions.

nuScenes [3] is a large dataset and contains 1,000 driving sequences in total. Among them, there are 700 scenes split for training, 150 scenes for validation, and 150 scenes for testing. It contains LiDAR, camera, and radar sources with a complete 360° environment. The main evaluation metrics are mean average precision (mAP) and nuScenes detection score (NDS).

### 4.2   Implement Details

SQDNet adopts an architecture similar to Voxel R-CNN [12] and develops SQDNet using the OpenPCDet open-source 3D object detection framework [30]. Tab. 1 details the parameters of the sparse 3D Backbone utilized in SQDNet. SQDNet follows the same training loss and dataset settings as Voxel R-CNN [12] and adopts widely used data augmentation techniques, including gt-sampling, rotation, flipping, scaling, and local noising. SQDNet pretrains the depth completion network on the KITTI dataset and fixes the parameters when training SQDNet.

The thresholds $\tau$, $\mu$, and $\sigma$ in SQD are set to 10, 3, and 0.9, respectively. Set the size of the 2D-occupied grid to 5, 76. Consequently, we derive $m$ and $n$ based on the depth value and depth map width, setting $m = \frac{d}{5}$ and $n = \frac{w}{76}$, where $d$ represents the maximum depth value observed in the depth map, and $w$ signifies the width of the depth map. SQDNet is trained on 2 RTX 3090 GPUs. Diverging from SFD [38] and VirConv [37], this paper employs dense depth maps generated by KBNet [34], chosen over alternatives like TWISE [18] and PENet [17] due to KBNet's demonstrated faster inference speed, as evidenced in Tab. 2.

## 4.3   Main Results

We report the Car 3D Detection results on the KITTI test set in Tab. 3. SQDNet outperforms the Voxel R-CNN [12] with improvements of 0.68%, 0.2%, and 2.01% in 3D AP(R40) for *easy*, *moderate* (*mod.*), and *hard* categories, respectively. Notably, for *hard* category detection, SQDNet achieves a score of 79.07%, surpassing several methods, including SFD [38], 3ONet [16], and PVT-SSD [43]. In Bird's Eye View (BEV) detection, SQDNet continues to excel, especially in *hard* categories as shown in Tab. 4, improving upon Voxel R-CNN [12] by margins of 0.59%, 1.80%, and 1.91%. SQDNet also achieves superior performance in *hard* category detection with a score of 88.04%, outdoing other methods such as LoGoNet [21] and TED [36]. The significant performance enhancement primarily stems from using pseudo points to fill in structural details on the object's surface.

## 4.4   Ablation Study

Here, we provide extensive experiments to analyze the effectiveness of SQDNet. Because SQD is based on 2D-occupied grids from a top-view perspective to sample pseudo points, we evaluate the performance of each module in Car BEV detection.

**Performance with different sparse 3D Backbone**   We replace the sparse 3D Backbone of voxel R-CNN [12] with Backbones proposed in other papers while maintaining consistency in training-related settings to compare the performance on the KITTI val fairly. Using the Voxel R-CNN implementation of OpenPCDet [30] as our baseline, we retrain the model. Our modified version, Voxel R-CNN†, incorporates pseudo points to enhance the LiDAR point cloud and adjusts the number of channels in the sparse 3D Backbone from [16, 32, 64, 64] to [16, 32, 64, 128].

The results indicate that SQDNet significantly enhances performance over Voxel R-CNN [12], with improvements of 1.19%, 0.42%, and 2.3% in the *easy*, *mod.*, and *hard* categories of Car BEV detection, respectively (see Tab. 5). This underscores the efficacy of supplementing LiDAR data with pseudo points to fill in structural information on object surfaces, thereby enhancing detection performance. By expanding the **kernel size** of the convolutional kernel, SQDNet broadens the receptive field of features and models long-range dependencies, avoiding frequent projection of 3D voxels into 2D space. SQDNet allows pseudo voxels to interact with accurate features from LiDAR voxels during the convolution process, thereby mitigating the impact of local structural ambiguity of pseudo points. The results demonstrate the effectiveness of SQDNet.

**Sampling time for different sampling methods**   Tab. 6 presents a comparison of sampling times between SQD and other point cloud sampling methods when applied to pseudo points. Notably, uniforml_down_sample, voxel_down_sample, and curvature_down_sample are methods derived from Open3D. Open3D is an open-source library that supports the rapid development of software that deals with 3D data [52].

During the training process, we calculated the average number of pseudo points queried by SQD for each LiDAR point cloud frame, which amounted to approximately 20,000 points. Consequently, we employ various sampling methods to sample 20,000 points from the original pseudo point cloud and compare their sampling times. By mapping both LiDAR and pseudo points onto 2D-occupied grid

**Table 1: The parameters of sparse 3D Backbone utilized in SQDNet.**

|  | Conv_input | Layer1 | Layer2 | Layer3 | Layer4 | Conv_out |
|---|---|---|---|---|---|---|
| Conv type | SubM Conv3D 

 in/out: 5/16 
 kernel_size: 3 
 padding: 1 | LargeKernel3D 

 in/out: 16/16 
 kernel_size: 7 
 padding: 2 | Sparse Conv3D 
 in/out: 16/32 
 kernel_size: 5 
 stride: 2 
 padding: 2 | Sparse Conv3D 
 in/out: 32/64 
 kernel_size: 5 
 stride: 2 
 padding: 2 | Sparse Conv3D 
 in/out: 64/128 
 kernel_size: 5 
 stride: 2 
 padding: (0,1,1) | Sparse Conv3D 
 in/out: 128/128 
 kernel_size: (3,1,1) 
 stride: (2,1,1) 
 padding: 0 |
| Conv type |  | LargeKernel3D 
 in/out: 16/16 
 kernel_size: 7 
 padding: 2 | SubM Conv3D 
 in/out: 32/32 
 kernel_size: 5 
 padding: 1 | SubM Conv3D 
 in/out: 64/64 
 kernel_size: 5 
 padding: 1 | SubM Conv3D 
 in/out: 128/128 
 kernel_size: 3 
 padding: 1 |  |
| Conv type |  |  | SubM Conv3D 
 in/out: 32/32 
 kernel_size: 5 
 padding: 1 | SubM Conv3D 
 in/out: 64/64 
 kernel_size: 5 
 padding: 1 | SubM Conv3D 
 in/out: 128/128 
 kernel_size: 3 
 padding: 1 |  |

**Table 2: Comparison of inference time for different depth completion models.**

| Method | Reference | Infer. time (s) |
|---|---|---|
| TWISE [18] | CVPR 2021 | 0.022 |
| PENet [17] | ICRA 2021 | 0.032 |
| KBNet [34] | ICCV 2021 | 0.016 |

**Table 3: Quantitative detection performance for Car 3D detection on the KITTI test.**

| Method | Reference | 3D Car AP(R40) | | |
|---|---|---|---|---|
| | | Easy | Mod. | Hard |
| Voxel R-CNN [12] | AAAI 2021 | 90.90 | 81.62 | 77.06 |
| PDV [19] | CVPR 2022 | 90.43 | 81.86 | 77.36 |
| SIENet [22] | PR 2022 | 88.22 | 81.71 | 77.22 |
| GLENet [50] | IJCV 2023 | 91.67 | 83.23 | 78.43 |
| Focals Conv [8] | CVPR 2022 | 90.55 | 82.28 | 77.59 |
| 3Onet [16] | IEEE SENS J 2023 | **92.03** | **85.47** | 78.64 |
| 3D HANet [39] | TGRS 2023 | 90.79 | 84.18 | 77.57 |
| SFD [38] | CVPR 2022 | 91.73 | 84.76 | 77.92 |
| PVT-SSD [43] | CVPR 2023 | 90.65 | 82.29 | 76.85 |
| FARP-Net [40] | TMM 2023 | 88.36 | 81.53 | 78.98 |
| Ada3D [51] | ICCV 2023 | 87.46 | 79.41 | 75.63 |
| SQD(Ours) | | 91.58 | 81.82 | **79.07** |

**Table 4: Quantitative detection performance for Car BEV detection on the KITTI test.**

| Method | Reference | BEV Car AP(R40) | | |
|---|---|---|---|---|
| | | Easy | Mod | Hard |
| Voxel R-CNN [12] | AAAI 2021 | 94.85 | 88.83 | 86.13 |
| GraR-Po [42] | ECCV 2022 | 95.79 | 92.12 | 87.11 |
| PA3DNet [31] | TII 2023 | 93.11 | 89.46 | 84.60 |
| GLENet [50] | IJCV 2023 | 93.48 | 89.76 | 84.89 |
| 3Onet [16] | IEEE SENS J 2023 | **95.87** | 90.07 | 85.09 |
| 3D HANet [39] | TGRS 2023 | 94.33 | 91.13 | 86.33 |
| LoGoNet [21] | CVPR 2023 | 95.48 | 91.52 | 87.09 |
| PVT-SSD [43] | CVPR 2023 | 95.23 | 91.63 | 86.43 |
| SFD [38] | CVPR 2022 | 95.64 | 91.85 | 86.83 |
| TED [36] | AAAI 2023 | 95.44 | **92.05** | 87.30 |
| SQD(Ours) | | 95.44 | 90.63 | **88.04** |

**Table 5: Quantitative detection performance with different sparse 3D Backbone on the KITTI val.**

| Conv | Parm | BEV Car AP(R40) | | |
|---|---|---|---|---|
| | | Easy | Mod. | Hard |
| Voxel R-CNN [12] | - | 95.68 | 91.25 | 88.95 |
| LargeKernel3D [9] | 13.7M | 95.54 | 88.71 | 88.36 |
| Focal Conv [8] | 9.5M | 95.46 | 88.37 | 87.79 |
| SPSS [23] | 9.2M | 95.89 | 88.41 | 86.04 |
| Voxel R-CNN † | 9.2M | 96.46 | 91.45 | 91.04 |
| VirConv [37] | 13M | 96.64 | 91.38 | 90.94 |
| SQD(Ours) | 11M | **96.87** | **91.67** | **91.25** |

maps and querying pseudo points based on the occupancy status of 2D grids and the density of LiDAR points within "occupied" grids, the fast sampling speed of SQD has been achieved.

**Ablation study on SQDNet** Tab. 6 presents an analysis of sampling times for various methods, demonstrating that SQD and random_sample fulfill the real-time requirements of SQDNet. To ascertain the efficacy of SQDNet's components, we conduct experiments as depicted in Tab. 7. Experiment (a) uses Voxel R-CNN [12] implemented by OpenPCDet [30] as the baseline. Experiment (b) utilizes the SQD to query pseudo points and the sparse 3D Backbone, Voxel R-CNN†. Experiment (c) applies random_sample to

sample 20,000 points, integrating them with LiDAR points to generate a new point cloud. We then employ our proposed sparse 3D Backbone to extract voxel features from this enriched point cloud. Experiment (d) is SQDNet. Results reveal that using SQD to query pseudo points effectively fills in structural information on object surfaces, significantly improving detection accuracy, especially for hard-to-detect objects.

**Table 6: Sampling time for different sampling methods.**

| Method | Time (s) |
|---|---|
| SQD | **0.01** |
| random_sample | 0.007 |
| uniform_down_sample | 0.24 |
| voxel_down_sample | 0.28 |
| farthest_point_sample | 1.95 |
| curvature_down_sample | 11.99 |

**Table 7: Ablation study on SQDNet. The results are calculated by 40 recall positions for Car BEV detection.**

| Experiment | SQD | 3D Backbone | BEV Car | | |
|---|---|---|---|---|---|
| | | | *Easy* | *Mod.* | *Hard* |
| (a) | | | 95.68 | 91.25 | 88.95 |
| (b) | ✓ | | 96.46 | 91.45 | 91.04 |
| (c) | | ✓ | 96.31 | 91.43 | 91.17 |
| (d) | ✓ | ✓ | **96.87** | **91.67** | **91.25** |

**Table 8: Performance on different distances.**

| With SQD | Distance | | |
|---|---|---|---|
| | 0-20m | 20-40m | 40m-inf |
| No | 92.91 | 81.00 | 28.83 |
| Yes | 93.32 | 82.38 | 29.93 |
| *Improvement* | +0.41 | +1.38 | +1.10 |

**Table 9: Inference speed of different methods.**

| **SQD** | **SFD** [38] | **3ONet** [16] | **TSSTDet** [15] |
|---|---|---|---|
| 13.5 FPS | 10.2 FPS | 6.5 FPS | 7.7 FPS |
| **TED** [36] | **CasA** [35] | **Voxel R-CNN** [12] | **GLENet** [50] |
| 11.1 FPS | 11.6 FPS | 21.08 FPS | 20.82 FPS |

**Conditional Analysis** To determine the scenarios where our method significantly improves the baseline, we evaluate SQDNet across varying distances. As illustrated in Tab. 8, the most significant improvements are observed with distant objects. This finding corroborates our hypothesis that integrating pseudo points with LiDAR point clouds significantly increases the detection accuracy for objects with sparse LiDAR points.

**Inference Speed** We test the inference speed of SQDNet on the NVIDIA RTX 3090 GPU. With the depth completion network, the speed of SFD is 13.5 FPS, as shown in Tab. 9.

**Evaluation on the nuScenes val set** To demonstrate our method's universality, we conduct experiments on the nuScenes [3], comparing our approach with CenterPoint [47], LargeKernel3D [9], and MVP [46]. Adopting MVP's data augmentation strategy, we train the network for 20 epochs on 4 RTX 3090 GPUs. The results are presented in Tab. 10. Given the absence of the depth completion task in the nuScenes, we utilize virtual points generated by MVP to enrich the LiDAR point cloud. Our proposed sparse 3D Backbone is then employed to extract voxel features, integrating with the modules in CenterPoint to fulfill the 3D object detection

task. Merging VP+SQDNet+CentrPoint increases the mAP by 0.2, but decreases the NDS by 0.2. This result may be due to the use of the LiDAR with 32 lasers in the nuScenes, which results in fewer LiDAR points and exacerbates the problem of local structural blurring in virtual points.

**Table 10: 3D detection results on the nuScenes val set.**

| Method | mAP | NDS |
|---|---|---|
| CenterPoint [47] | 56.4 | 64.8 |
| LargeKernel3D [9] | 63.3 | 69.1 |
| MVP [46] | 66.0 | **69.9** |
| VP + SQDNet + CenterPoint | **66.2** | 69.7 |

## 4.5 Qualitative and quantitative analysis of SQD

Fig. 7 illustrates the average numbers of LiDAR and pseudo points within GTboxes across different distances, highlighting a notable difference between the numbers of pseudo and LiDAR points. By using SQD to query pseudo points to supplement the LiDAR point cloud, there's a significant increase in point density inside GTboxes.

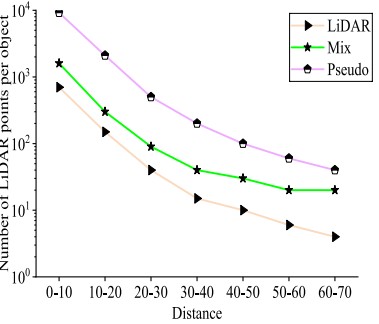

**Figure 7: The average numbers of LiDAR points, pseudo points, and mixed points obtained by using SQD within GT-boxes at different distances. It can be seen that using SQD to query pseudo points can supplement the LiDAR points within GTboxes.**

In Fig. 8, we show the LiDAR points and their mixed points of six Cars, where the mixed points are a mixture of pseudo points queried through SQD and LiDAR points. The IDs assigned to these Cars correspond to the IDs in the accompanying top RGB image. Cars are arranged in order of detection difficulty from left to right: *hard*, *mod.* and *easy*.

This figure vividly demonstrates that pseudo points, queried by SQD, effectively enrich sparse areas within the LiDAR point clouds for objects categorized as *hard*. However, for objects like Car2 that are obscured in the middle, SQD cannot adequately fill the gaps in the LiDAR point cloud. The top view of Car2's LiDAR and mixed point clouds reveals two visible blank spaces. This is due to the fact that objects occluded in the RGB image remain occluded in the corresponding dense depth map. This partially explains SQDNet's limited improvement in detecting objects of *mod.* category.

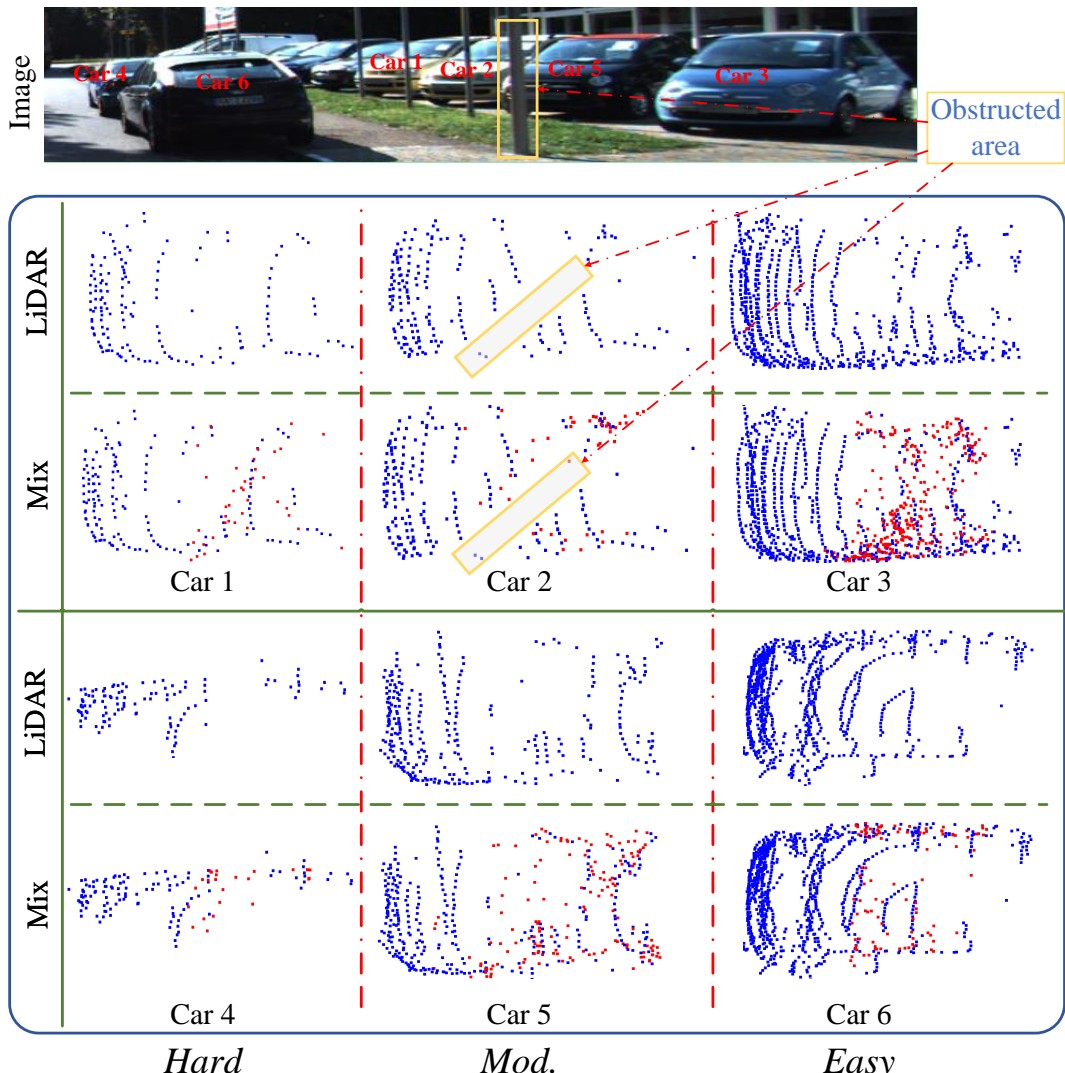

**Figure 8: Qualitative analysis of LiDAR and mixed points for six cars. This figure illustrates the comparison between original LiDAR points (blue dots) and mixed points (red dots), the latter being a blend of LiDAR points and pseudo points. Cars' IDs match those in the corresponding RGB image, with cars arranged from left to right by detection difficulty: *hard, mod., easy.***

## 5  CONCLUSION

In this paper, we present SQDNet, a novel 3D object detection framework that utilizes the SQD for rapid sampling of large-scale pseudo points on the dense depth map directly, significantly enhancing the structural detail on object surfaces. Additionally, we develop a sparse 3D Backbone designed to broaden the receptive field of features. This design enables the effective integration of accurate features from LiDAR voxels into pseudo voxels during the 3D convolution process. Our experiments demonstrate the effectiveness of the SQDNet, particularly in detecting difficult-to-detect instances.

Despite its successes, SQDNet faces several limitations, such as the need for manual tuning of 2D-occupied grid sizes and density thresholds, which proves to be labor-intensive. Second, although SQDNet shows considerable performance enhancements in the *easy*

and *hard* categories of the KITTI 3D object detection benchmark, improvements in the *mod.* category are less pronounced.

Future efforts will focus on simplifying SQD to adaptively query pseudo points with minimal to no manual parameter tuning. We've already demonstrated the effectiveness of using pseudo points to enrich sparse LiDAR point clouds in car detection and aim to apply this approach to other categories, like bicycles and pedestrians, to enhance LiDAR data in diverse scenarios. Our qualitative analysis in Fig. 8 reveals that SQDNet shows a slight improvement in detecting objects of *mod.* category, which may be due to its ineffectiveness in supplementing pseudo points for occluded objects, which is a challenge we need to address in future research.

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
