# OpenReview forum: "Sparse Query Dense: Enhancing 3D Object Detection with Pseudo points"
_acmmm.org/ACMMM/2024/Conference — MM2024 Oral_

### Official Review · Reviewer_Z67Q · 2024-05-21

**Rating:** 3
**Confidence:** 3

**Summary:**

In summary, thish paper proposed a 3D object detection framework, which utilizes the SQD for rapid sampling of large-scale pseudo points on the dense depth map directly to enhance the structural detail on object surfaces.

**Strengths:**

- Outperforms existing solutions in specific categories of metrics.
- This article is somewhat innovative, introducing depth completion and occupancy grid;

**Limitations:**

There are some key points and suggestions that the authors should address:

- It is inaccurate that the projection in the title of Fig2 & Fig3 only requires external parameters. Projecting the point cloud to the UV plane also requires internal camera parameters;
- What is the difference between Table1 and Fig6? This table is input and output from top to bottom and left to right? It is not easy to understand. It is recommended to combine Fig6 and Table1 to make a more detailed picture for display;
- In Fig 2, what is the input of the camera rgb image used for? There seems to be no clear explanation. I guess it is used as input for the depth completion scheme of KBNet?
- Why is the deep completion solution shown in Table 2 stuck in 2021? Is there no better performance option for the new solution in 2022-2024?
- Table 8 It is recommended to indicate what indicator is reported. I guess it should be AP;
- Are the inference speeds of other solutions in Table 9 from the reports of their respective papers, or are they also tested on the 3090? Are the hardware platforms unified?
- The scaling of the Fig 7 picture is abnormal;
- What is the main reason for the lack of significant improvement in the moderate category?
- Typos, etc., such as L346 upper and lower case, L570 uniforml, Mod. Mod in Table3/4/5 recommends a unified format, etc.

**Suitability:**

2

---

### Official Review · Reviewer_hn9C · 2024-05-23

**Rating:** 4
**Confidence:** 3

**Summary:**

The paper titled "Sparse Query Dense: Enhancing 3D Object Detection with Pseudo points" introduces SQDNet, a novel framework aimed at improving 3D object detection in LiDAR-based systems. It addresses the challenge of sparse point clouds by utilizing a sparse-to-dense matching approach, facilitated through grid position indices to streamline the sampling of large-scale pseudo points from dense depth maps. This technique enhances the point cloud data, especially useful in autonomous driving applications.

**Strengths:**

The SQDNet framework is innovative, particularly in its use of grid-based sparse-to-dense matching to enhance the quality of pseudo points in LiDAR data. This method not only optimizes the data preprocessing pipeline by enabling rapid, large-scale sampling but also alleviates issues of uneven distribution and noise in pseudo points through the integration of a sparse 3D backbone.

The authors thoroughly evaluate the framework on the KITTI benchmark, demonstrating significant improvements in object detection accuracy, particularly for hard-to-detect objects. These results validate the effectiveness of their approach.

The paper is well-organized, with clear explanations of the methodology and its components. The use of figures and tables effectively illustrates the processes and benefits of the proposed framework.

**Limitations:**

1. The necessity for manual tuning of grid sizes and density thresholds might limit the practicality and scalability of the proposed method in diverse operational environments.

2. While the paper shows considerable improvements in detecting hard objects, the gains in easy and moderate scenarios are less pronounced, which may impact its effectiveness in balanced real-world conditions.

**Suitability:**

2

---

### Official Review · Reviewer_gE9b · 2024-05-25

**Rating:** 4
**Confidence:** 3

**Summary:**

Addressing the issues of inaccurate depth generation and complex sampling processes in existing point cloud-based methods, this paper proposes a novel approach, SQDNet, which primarily includes the SQD module to alleviate these problems. The effectiveness of SQD has been validated through improvements on the KITTI dataset at the hard difficulty level.

**Strengths:**

1.By mapping sparse and dense points to 2D-occupied grid maps, we facilitate rapid sparse-to-dense matching leveraging grid position indices and reduce the computational cost of using LiDAR points to query pseudo points

2.SQD samples pseudo points based on the occupancy status of grids and density of LiDAR points within these grids to alleviate edge noise and the problem of uneven distribution;

3.A novel sparse 3D Backbone is designed to model long-range dependencies, alleviating the problem of local structural ambiguity;

4.Experimental results demonstrate that using pseudo points to supplement surface structure details of difficult-to-detect objects significantly enhances the model’s ability to detect these objects.

**Limitations:**

1.In Section 3.1, Sparse Query Dense, and Figure 4, the paper generates dense pseudo point clouds by querying from a sparse occ map. How is this querying implemented? Is it done through the transformer's query mechanism? Please provide the specific implementation process.

2.The sparse 3D backbone in Section 3.2 is almost identical to the 3D backbone in VirConv, except that the kernel size is larger. The 3D backbone part in Figure 2 is also the same as that in VirConv. Please explain the differences between your approach and VirConv, and how it alleviates the issue of local blurring in the pseudo point cloud structure (the problem described in Figure 1f).

3.The edge noise problem described in Figure 1 does not seem to affect the final detection performance. This noise is caused by inaccurate depth, which appears to be unrelated to the improvement in sampling speed discussed in Section 3.1. Please clarify the relationship between the problem in Figure 1 and the method implemented in Section 3.1.

4.The distribution of the pseudo point cloud (red points) generated in Figure 5 appears to be different from the original point cloud (blue points), resembling the irregular distribution described in Figure 1b. This implies that the pseudo point cloud generated using the SQD module does not resolve the problem described in Figure 1. The relationship between the motivation in Figure 1 and the modules of SQDNet needs further clarification

5.In the experiments, Table 2 seems redundant. Comparing the inference times of different depth completion networks is meaningless (suggest removing it). Figures 3 and 4 both show results on the test set, and it is recommended to combine them. The last two rows in Figure 8 also seem redundant. Is the obstructed area corresponding to the BEV view in the figure? If so, it might cause ambiguity in the mode difficulty level for cars, as occlusion is not an issue in the BEV view (one of the reasons BEV methods are popular). It is unreasonable to attribute the small improvement of the detector at the mode difficulty level to occlusion in the paper. The specific reasons need further clarification.

6.It is recommended to compare your approach with the VirConv detector or integrate the SQDNet module into the VirConv detector to achieve better detection results.

**Suitability:**

2

---

### Meta-Review · Area_Chair_7RHd · 2024-07-03

**Recommendation:** Accept (Oral)
**Confidence:** 3

**Metareview:**

This paper deals with 3D object detection via LiDAR point clouds. According to the 3 confident reviewers, the contribution is technically sound and somewhat innovative. It outperforms SOTA, but only in specific categories of metrics (hard difficulty level), with less pronounced gains (or no gain at all) in easy/moderate scenarios, which questions its effectiveness and robustness in real-world conditions. Added to the unimodal nature of the data and application involved, this weakness makes the contribution questionable for the ACM MM community.